# Mangiferin-Loaded Smart Gels for HSV-1 Treatment

**DOI:** 10.3390/pharmaceutics13091323

**Published:** 2021-08-24

**Authors:** Mariaconcetta Sicurella, Maddalena Sguizzato, Rita Cortesi, Nicolas Huang, Fanny Simelière, Leda Montesi, Peggy Marconi, Elisabetta Esposito

**Affiliations:** 1Department of Chemical, Pharmaceutical and Agricultural Sciences, University of Ferrara, I-44121 Ferrara, Italy; scrmcn@unife.it (M.S.); sgzmdl@unife.it (M.S.); crt@unife.it (R.C.); 2CNRS, Institut Galien Paris-Saclay, Université Paris-Saclay, 92296 Châtenay-Malabry, France; nicolas.huang@universite-paris-saclay.fr (N.H.); fanny.simeliere@universite-paris-saclay.fr (F.S.); 3Cosmetology Center, University of Ferrara, I-44121 Ferrara, Italy; cosm@unife.it

**Keywords:** phosphatidylcholine, Pluronic organogel, in vitro diffusion, mangiferin, HSV-1

## Abstract

Infections due to HSV-1 affect many people all over the world. To counteract this pathology, usually characterized by perioral sores or by less frequent serious symptoms including keratitis, synthetic antiviral drugs are employed, such as acyclovir, often resulting in resistant viral strains under long-term use. Many plant-derived compounds, such as mangiferin and quercetin, have demonstrated antiviral potentials. In this study, smart semisolid forms based on phosphatidylcholine and Pluronic were investigated as delivery systems to administer mangiferin on skin and mucosae affected by HSV-1 infection. Particularly, lecithin organogels, Pluronic gel, and Pluronic lecithin organogels were formulated and characterized. After the selection of gel compositions, physical aspects, such as rheological behavior, spreadability, leakage, and adhesion were evaluated, suggesting a scarce suitability of the lecithin organogel for topical administration. Mangiferin was efficiently included in all type of gels. An in vitro study based on the Franz cell enabled us to find evidence of the gel capability to control drug diffusion, especially in the case of Pluronic organogel, while an in vivo study conducted on human volunteers demonstrated the safeness of all of the gels after cutaneous administration. Furthermore, a plaque reduction assay demonstrated the virucidal effect of mangiferin loaded in a Pluronic gel and a Pluronic lecithin organogel against the HSV-1 KOS strain.

## 1. Introduction

Herpes simplex virus type 1 (HSV-1) is a common human pathogen affecting billions of people worldwide. HSV-1 infection, mainly transmitted by oral contact, is characterized by the formation of epidermal lesions inside and around the mouth as well as in the nose or in other body zones, such as the fingers, with frequent recurrences [1,2,3]. In some cases, for instance in immunocompromised patients, HSV-1 infection can cause severe complications, including encephalitis or keratitis [4]. To treat HSV-1 infections, many antiviral drugs are in clinical use, among which acyclovir is the first-line drug [5]. Despite its efficacy, the long-term use of acyclovir induces drug-resistant viral strains; therefore, new antiviral alternative drugs against drug-resistant HSV-1 are strongly required [6,7]. In this respect many plant extracts widely used in medicine possess interesting therapeutic potential, such as antimicrobial and antiviral activities. For instance, balm mint extracts, Brazilian propolis extracts, rosemary, lemon balm oil, quercetin, and mangiferin (MG) demonstrated and inhibitory activity against HSV-1 [8,9,10,11,12].

MG is a natural glucosyl xanthone derived from mango or papaya, characterized by an interesting pharmacological potential, exerting antioxidant, antiapoptotic, anti-inflammatory, anticarcinogenic, and antidiabetic properties as well as antiviral action against herpes simplex virus and polio virus [12,13,14,15]. A recent investigation has highlighted the MG virucidal effect against HSV-1, suggesting its topical use to inactive virus replication and to minimize the selection of resistance, thus counteracting acyclovir-resistant infections [12]. Notwithstanding its therapeutic potential, MG is not soluble in water and has low bioavailability, requiring the use of specialized delivery systems able to dissolve the drug while maintaining its stability against oxidative degradation [16].

Particularly, in order to deliver MG on the skin and mucosa affected by herpetic infection, semisolid forms characterized by biocompatibility and permeation enhancement properties are needed. In this respect, smart gels constituted of phosphatidylcholine (PC) and/or Pluronic copolymers appear to be an appropriate strategy. Indeed, PC is a natural biocompatible surfactant able to self-organize in water and/or organic solvents, giving rise to many supra-molecular structures and to increased skin permeation due to its affinity with skin lipids [17,18]. When PC is solubilized in biocompatible organic solvents, it forms reverse micelles that, under the addition of precise amounts of water, longitudinally grow, resulting in an entangled three-dimensional network of reverse cylindrical micelles [19]. The viscosity of these so-called lecithin organogels (LO) is strongly influenced by the amount of added water, resulting in oleaginous semisolid forms suitable for topical administration [20]. Due to their peculiar structure, LO can dissolve both hydrophilic and lipophilic drugs, while promoting their release under transdermal or transmucosal delivery [18,21].

The non-ionic poly(oxyethylene)poly(oxypropylene) (PEO-PPO) block copolymer Pluronics behave as surfactants. Indeed, Pluronics can self-assemble in water-forming micelles, suitable for lipophilic molecule solubilization. Depending on molecular weight and POP percentage, some Pluronics such as F127 in water display a thermo-reversible character [22]. Indeed, F127 solution (20–30%, p/p) above a T_sol-gel_ transition temperature (20–30 °C) undergoes a sol–gel transition phenomenon, resulting in a stiff gel due to a close micelle packing [23]. The smart behavior of Pluronic gels (PGs) properly fits cutaneous and mucosal administration, being easily administrable as a liquid micellar solution while becoming viscous when approaching body temperature, forming semisolid forms able to control drug release [24]. A peculiar smart semisolid formulation can be typified by Pluronic lecithin organogel (PLO) as a combination of LO and PG. Indeed, PLO are formed by mixing PG and an organic solution of PC, thus introducing F127 in the aqueous phase of LO [25]. The architecture of PLO is complicate, being a biphasic system where both PC and F127 in the presence of an organic solvent and water self-organize into microstructures that intertwine with each other [26]. Both PC and PPO blocks in F127 can promote the permeation of associated drugs, interfering with the bilayer lipid packing of the cellular membrane [27]. Notwithstanding a few PLO-based medicines available on the market, PLO’s potential use in the treatment of many diseases is under preclinical or clinical investigations [28]. Notably, since LO, PG, and PLO can be produced by cold methods, they appear particularly suitable for the accommodation and delivery of thermolabile drugs, preventing chemical degradation. Moreover, their preparation can be easily scaled-up for industrial production. In a recent study, the potential of acyclovir loaded in PLO was investigated on an HSV-1 infection model, demonstrating a major efficacy with respect to a marketed formulation [29].

In the present investigation, smart semi-solid forms suitable for MG topical application on mucosae and skin, such as LO, PG, and PLO, were designed and selected in order to treat HSV-1 infections. Namely, special regard has been devoted to studying some properties affecting topical application, such as rheological behavior, spreadability, leakage, and adhesion. Diffusion tests were conducted using Franz cells to investigate the influence of the vehicles on drug diffusion after the application of the semisolid forms, while an in vivo irritation test was carried out to verify their safeness.

Furthermore, the antiviral activity of MG-containing lecithin organogels, Pluronic gel, and Pluronic lecithin organogels was studied, evaluating their inhibitive capacity on the plaque formation of HSV-1 in cultured cells.

## 2. Materials and Methods

### 2.1. Materials

Mangiferin (MG), the copolymer poly(ethylene glycol)-block-poly(propylene glycol)-block-poly(ethylene glycol) poloxamer 407 (Pluronic F127, F127) (PEO98-POP67-PEO98), isopropyl palmitate, acid yellow 25, Coomassie brilliant blue G 250, and oil red O were purchased from Sigma-Aldrich (St Louis, MO, USA). The soybean lecithin (PC) (90% phosphatidylcholine) was Epikuron 200 from Lucas Meyer (Hamburg, Germany). Polytetrafluoroethylene membranes (2 cm diameter, pore size 0.2 μm) were purchased from Merck (Milan, Italy). The solvents were of HPLC grade, and all other chemicals were of analytical grade.

### 2.2. Lecithin Organogel Preparation

Lecithin organogel (LO) was prepared by adding precise amounts of water to 200 mM of the PC solution in isopropyl palmitate (PC/IPP) under magnetic stirring (700 rpm) at 20–25 °C. The amount of added water was expressed as the molar ratio of water to PC ([water]/[PC]) [19,30]. The maximum [Water]/[PC] was determined, i.e., the highest amount of water that can be incorporated into the PC solution without phase separation. The samples were maintained under stirring at 25 °C for 30 min and later examined macroscopically, detecting transparency and phase separation phenomena. In the case of MG-containing organogels (LO-MG), MG was solubilized in the PC/IPP solution before the addition of water. 

### 2.3. Pluronic Gel Preparation 

Pluronic gel (PG) was prepared according to the “cold technique” [31]. In brief, a weighed amount of F-127 was gradually added to cold water (5–10 °C) in an ice bath under magnetic stirring (500 rpm) up to a final concentration of 20% *w*/*w*. The container was sealed and left overnight at 5 °C in a refrigerator. In the case of MG-containing gels (PG-MG), MG was added into the preformed gel and solubilized by stirring.

### 2.4. Pluronic Lecithin Organogel Preparation

For Pluronic lecithin organogel (PLO) preparation, a lipid phase constituted of PC/IPP and an aqueous phase constituted of PG were previously prepared. To prepare PLO, different amounts of PC/IPP and PG were mixed under stirring (500 rpm) at 20–5 °C, resulting in 28:72 (PLO-7), 30:70 (PLO-3, PLO-4, and PLO-5), 33:66 (PLO-6), 50:50 (PLO-2), or 66:33 (PLO-1) *v*/*v* ratios between PC/IPP (O phase) and PG (W phase). In the case with MG-containing PLOs (PLO-MG), MG was solubilized in PG before addition into PC/IPP. 

### 2.5. Rheological Measurements

Rheological measurements of LO, PG, PLO-3, and PLO-7 were performed at 20 or 35 °C with an AR-G2 controlled-stress rotational rheometer (TA Instruments, New Castle, DE, USA) [32]. The geometry used was either a titanium cone-plate (diameter 40 mm, angle 1°, and truncation 28 µm) or a plate-plate (diameter 20 or 40 mm) to adapt the geometry to the sample consistency and available volume. A solvent trap was used to prevent solvent evaporation during the experiments. Shear rate sweeps (from 1000 to 0.01 s^−1^ during 3 min) were performed in order to obtain the corresponding viscosity curves. Temperature ramps were taken from 5 °C to 50 °C with a temperature rate 1 °C/min, controlled by a Peltier plate. A 2 min conditioning time at 5 °C was applied before starting the experiments. Measurements were performed thrice at least for each sample to ensure reproducibility.

#### Determination of Flow Properties 

The consistency and flow indices were determined from the power law described in Equation (1) for quantitative analysis of the flow behavior: *τ* = *K*·*γ^n^*
(1)
where *τ* is the shear stress (Pa), *γ* is the shear rate (s^−1^), *K* is the consistency index (Pa × s*^n^*), and n is the flow index (dimensionless) [33].

### 2.6. Spreadability Studies

The spreading capacity of LO, PG, and PLO was evaluated at ambient temperature (25 °C) 24 h after gel preparation [34]. Precisely, 100 mg of gel was placed in the center of a Petri dish (3 cm diameter) and then subjected to pressure by a glass dish carrying a 50 g mass. The diameter of the area occupied by the formulation in a predetermined time (10 s) was measured. The spreadability test was performed three times, and the mean values ± standard deviations were calculated using the following equation:*S* = *m* × *l*/*t*(2)
where *S* is the spreadability of the gel formulation, *m* is the weight (g) tied on the upper plate, *t* is the time (10 s), and *l* is the diameter (cm) of the area occupied by the gel in 10 s under pressure [34].

### 2.7. Leakage and Adhesion Tests 

For leakage and adhesion tests, the semisolid forms LO, PG, and PLO-3 were colored. Particularly in the case of LO, oil red (0.05% *w*/*w*) was dissolved in PC/IPP before water addition, and to color PG, acid yellow 25 (0.05% *w*/*w*) was directly dissolved in the F127 aqueous solution, while in the case of PLO-3 Coomassie blue (0.05% *w*/*w*) was dissolved in the F127 aqueous solution before addition into PC/IPP.

To test the in vitro leakage and adhesion of the formulations, agar (1.5% *w*/*w*) was added to a phosphate buffer with pH 7.4 and stirred at 95 °C until solubilization. After cooling, the obtained gels were cut into rectangular agar slides. For the leakage test, 50 mg of colored formulations were placed onto one end of an agar slide placed in a Petri plate. The Petri plate was vertically put at an angle of 90° on a transparent box wall and maintained at 32–35 °C. The running distance along the slide was measured 10 s after the gel placement. Gel leakage, expressed as the distance traveled by the gel, was measured three times, and the mean values ± standard deviations were calculated.

For the in vitro adhesion test, 50 mg of colored LO, PG, and PLO-3 were placed at the center of the agar slides. The agar slides were immersed in 10 mL of phosphate buffer pH 7.4 at 37 ± 1 °C for 2 h. The formulation residence times on the slides were visually compared after 10 and 60 min, measuring the increase in the area occupied by the gels using ImageJ software [34]. The tests were performed three times.

An in vivo skin adhesion test was performed on a volunteer, applying 30–40 mg of colored LO, PLO-3, and PG on the inner part of the forearm rotated outward. Forearm images were taken immediately after gel application, and 1 min or 10 min after rotating the forearm inward in vertical position. 

### 2.8. Evaluation of MG Content of Semisolid Forms

The content of MG within the gels was determined 1 and 30 days after preparation, diluting the gels with ethanol in a 1:10 *w*/*w* ratio. After magnetic stirring for 30 min and filtration of the solution by nylon syringe filters (0.22 μm pores), the MG amount was analyzed by HPLC as reported below. The physical stability of the gels was visually assessed on samples kept at room temperature in the light for 3 months.

### 2.9. In Vitro Diffusion Experiments

In vitro MG diffusion was investigated by Franz cells assembled with polytetrafluoroethylene membranes (Vetrotecnica, Padova, Italy) [31]. Franz cells were set up with a lower receptor and an upper donor compartment separated by a membrane, of which the exposed surface area was 0.78 cm^2^ (1 cm diameter orifice). Before starting the experiment, the membranes were hydrated with the receiving phase for 1 h. Five milliliters of ethanol/water (50:50, *v*/*v*) were poured in the lower section, stirred at 500 rpm by a magnetic bar, and maintained at 35 ± 1 °C using a thermostat during all the experiments [35] (FDA Guidance). Approximately 1 g of MG containing semisolid forms or 1 mL of the ethanol/water (30:70, *v*/*v*) solution of MG (MG 0.7 mg/mL) was placed in the donor compartment and sealed to avoid evaporation. At predetermined time intervals (0.5–7 h), 200 μL of the receiving phase was withdrawn, replaced with an equal volume, and analyzed for MG content by HPLC. The MG concentrations were determined six times in independent experiments, and the mean values ± standard deviations were calculated. The accumulation curves were obtained by plotting mean values as a function of time. The fluxes were extrapolated from the linear portion of the curves considering the slopes of the regression line (angular coefficient). Lastly, the diffusion coefficients were calculated according to Equation (3).
D = F/[MG](3)
where D is the diffusion coefficient; F is the flux; and [MG] is the MG concentration in the analyzed form, expressed in mg/mL. 

### 2.10. HPLC Analysis

HPLC analyses were conducted using a two-plungers alternative pump (Agilent Technologies 1200 series, Santa Clara, CA, USA), an UV-detector operating at 254 nm, and a 7125 Rheodyne injection valve with a 50 μL loop. A stainless-steel C-18 reverse-phase column (15 × 0.46 cm) packed with 5 μm particles (Platinum C18, Apex Scientific, Alltech, Lexington, KY, USA) was eluted with a mobile phase containing methanol/water 60:40 *v*/*v*, pH 4.0, at a flow rate of 1 mL/min. The method was validated for linearity (R^2^ = 0.994), repeatability (relative standard deviation < 0.05%, n = 6 injections), and limit of quantification (0.05 μg/mL).

### 2.11. Patch Test

An in vivo irritation test was performed in order to evaluate the effect of LO, PG, and PLO-3 applied in a single dose on intact human skin. The occlusive patch test was conducted at the Cosmetology Center of the University of Ferrara following the basic criteria of the protocols for skin compatibility testing of potentially cutaneous irritant cosmetic ingredients on human volunteers (SCCNFP/0245/99) [36,37,38]. The protocol was approved by the Ethics Committee of the University of Ferrara, Italy (study number: 170583). The test was run on 20 healthy volunteers of both sexes who gave written consent to the experimentation, excluding subjects affected by dermatitis, with a history of allergic skin reaction or under anti-inflammatory drug therapy (either steroidal or non-steroidal). Ten milligrams of LO-MG, PG-MG, and PLO-MG were posed into aluminum Finn chambers (Bracco, Milan, Italy) and applied onto the skin of the forearm or the back protected by self-sticking tape. Particularly, samples were directly applied into the Finn chamber by an insulin syringe and left in contact with the skin surface for 48 h. Skin irritative reactions (erythematous and/or edematous) were evaluated 15 min and 24 h after removing the patch and cleaning the skin from sample residual. Erythematous reactions have been sorted out into three groups according to the reaction degree: light, clearly visible, and moderate/serious erythema. The average irritation index was calculated as the sum of the erythema and edema scores and expressed according to a scale considering 0.5 as the threshold above which the product is to be classified as slightly irritating, 2.5–5 as moderately irritating, and 5–8 as highly irritating.

### 2.12. Antiviral Activity Study against HSV-1

#### 2.12.1. Cell Culture

Vero (African green monkey kidney, ATCC- LGC Standards S.r.l., Milan, Italy) cells were cultivated in Eagle’s minimum essential medium (DMEM); supplemented with 5% fetal bovine serum (FBS), 100 mg/mL penicillin, and 100 mg/mL streptomycin; and incubated at 37°C under 5% CO_2_ in an incubator. The cells were seed at 5 × 10^5^ per well in a six- well plate, 24 h prior to plaque assay [39]. 

#### 2.12.2. Herpes Virus Stock Generation

Vero cells (2 × 10^7^) in 10–20 mL of cell culture medium were seeded into a 175 cm^2^ tissue culture flask and incubate overnight at 37 °C in a humidified 95% air–5% CO_2_ incubator. Vero cells were infected with the herpes simplex virus strain KOS at a multiplicity of infection (M.O.I) of 0.01. The cells were incubated for 1 h at 37 °C to allow for adsorption of the virus into the cells. The flasks were rocked every 15 min in order to evenly distribute the inoculum. The virus inoculum was aspirated, and the cell culture medium was added for a final volume of 10–20 mL per flask. Infected cells were incubated at 37 °C in a humidified 95% air –5% CO_2_ incubator for 36–48 h until a complete cytopathic effect (CPE) was reached. The cells and supernatant were collected, the cells were removed by low-speed centrifugation, and the supernatant was centrifugated a 20,000 rpm for 30 min. The obtained pellet was resuspended in 1 mL of medium, aliquoted, and kept at −80 °C until use [39].

#### 2.12.3. Titration of Virus by Plaque Assay

The viral preparation was titrated on Vero cells. One day prior to titration, 6-well tissue culture plates with 0.5 × 10^6^ Vero cells per well were prepared. The virus was thawed on ice and sonicated for a few seconds prior to infection in order to separate virus particles. A series of ten-fold dilutions (10^−2^–10^−10^) of the virus stock in 1 mL of the cell culture medium without serum in 1.5 mL Eppendorf tubes were prepared and added to each six-well containing cells. The cells were incubated for 1 h at 37 °C to allow for adsorption of the virus into the cells. After 1 h of infection, the viral inoculum was removed and the monolayer was overlayed with 3 mL of 1% methylcellulose (Sigma). The plates were incubated for 3–4 days until well-defined plaques were visible. The methylcellulose medium was removed from the wells and stained for 10–20 min with 2 mL of crystal violet staining solution to fix the cells and the virus. The number of plaques was counted, and the average for each dilution (n = 3) was determined and multiplied by 10 to the power of the dilution to obtain the number of plaque forming units per ml (PFU/mL) [39]. 

#### 2.12.4. Antiviral Assay

The inhibition of virus replication (HSV-1 strain KOS 1 × 10^5^ pfu/mL) was measured by plaque assay, evaluating the MG-containing formulations using different protocols. Particularly, the antiviral assay was based on the pre-treatment, the infection treatment, the post-infection treatment, and the virucidal assay. 

#### 2.12.5. Pre-Treatment

Vero cells were pre-treated with SOL-MG (50 µg/mL), PG-MG (50 µg), PLO-3-MG (50 µg/mL), and LO-MG (50 µg/mL) for 1 h at 35 °C. The wells were washed and infected with different concentration of an HSV-1 KOS strain from a range of 1 × 10^5^ PFU/mL to 1 × 10 PFU/mL for 1 h to allow for adsorption of the virus to the cells. After 1 h at 35 °C, the viral inoculum was removed and 1% methylcellulose medium was added. The number of plaques was determined as previously described [39,40].

#### 2.12.6. Infection Treatment 

Vero cells were treated with SOL-MG (50 µg/mL), PG-MG (50 µg/mL), PLO-3-MG (50 µg/mL), and LO-MG (50 µg/mL) and simultaneously infected with the HSV-1 KOS strain from a range of 1 × 10^5^ PFU/mL to 1 × 10 PFU/mL for 1 h to allow for adsorption of the virus into the cells. After 1 h at 35 °C, the viral inoculum was removed and the cells were washed to remove viruses not adsorbed. The methylcellulose medium (1%) was added. The number of plaques was determined as previously described [39], and the percentage of viral inhibition was calculated by comparison with viral control.

#### 2.12.7. Post-Infection Treatment 

Vero cell monolayers were infected with different concentrations of the HSV-1 KOS strain from a range of 1 × 10^5^ PFU/mL to 1 × 10 PFU/mL for 1 h at 35 °C. After the adsorption time, the viral inoculum was removed and SOL-MG (50 µg/mL), PG-MG (50 µg/mL), PLO-3-MG (50 µg/mL), and LO-MG (50 µg/mL) were added for 1 h at 35 °C. The formulations were removed, and 1% methylcellulose medium was added. The number of plaques was determined as previously described [40], and the percentage of viral inhibition was calculated by comparison with viral control. 

#### 2.12.8. Virucidal Assay

HSV-1 KOS (1 × 10^5^ PFU/mL) was incubated with SOL-MG (50 µg/mL), PG-MG (50 µg), PLO-3-MG (50 µg/mL), and LO-MG (50 µg/mL), for 1 h and 6 h at 35 °C before cell infection. Ten-fold dilutions of the mixture or virus alone were adsorbed into the cells for 1 h at 35 °C. After the adsorption time, the viral inoculum was removed and the medium containing 1% methylcellulose was added. Plates were incubated for 3–4 days at 35 °C, washed, and fixed with crystal-violet to determine the reduction in the number of plaques in the treated virus compared with that in viral control [41]. 

#### 2.12.9. Statistical Analysis

All experiments were repeat three times, and statistical values were expressed as the mean ± standard deviation (SD). For all data analysis, GraphPad Prism 9 software (GraphPad Software Inc., San Diego, CA, USA) was used. Values of *p* < 0.05 were considered statistically significant.

## 3. Results and Discussion

A pre-formulative study was conducted to produce a semisolid form suitable for MG application on mucosae and skin. The formulation should (i) solubilize MG (oil/water partition coefficient 0.53) [16,42]; (ii) be easily applied on the cutaneous surface, localizing at a specific site; (iii) achieve a prolonged release of MG; and (iv) possibly improve its permeability [43]. In this regard, PC and F127 were chosen, characterized by an amphiphilic surfactant character and a transdermal potential. Notably, different types of formulations were considered, namely a lecithin organogel (LO), a poloxamer F127 hydrogel (PG), and a Pluronic lecithin organogel (PLO) as a hybrid form of the first two.

### 3.1. Preparation of Semisolid Formulations 

LO was prepared by adding trace amounts of water to a PC solution in IPP, selected as biocompatible excipients suitable for transdermal delivery. The [water]/[PC] molar ratio ranged between 1:1 and 3.5:1 [19]. After 10 min of magnetic stirring, the obtained gels appeared transparent, yellow, monophasic, and characterized by a viscosity proportional to the amount of added water [19,30]. Indeed, in LO, PC in IPP forms spherical or ellipsoidal reverse micelles that, upon successive addition of water, grow linearly, building a three-dimensional network of entangled polymer-like micelles, stabilized by hydrogen bonds between PC and water [44]. The maximum [water]/[PC] ratio selected for LO production, resulting in the thickest gel devoid from separation phenomena, was 3.5:1. 

PG production relied on the solubilization of F127 (20% *w*/*w*) in cold water under stirring, as previously reported [31]. In water, F127 exhibits a thermo-reversible behavior. Indeed, at low temperature, it self-aggregates into micelles, with the core mainly constituted of POP blocks and the corona of POE ones, while over its transition temperature (T_sol-gel_), F127 is organized in an ordered three-dimensional lattice [45]. Indeed, the heating of the F127 aqueous solution induces dehydration of the hydrophobic POP blocks and hydration of the hydrophilic POE blocks, leading to spherical micellular formation, organized in a paracrystalline packing [46,47]. The concentration of F127 was chosen on the basis of its T_sol-gel_ evaluated in a previous study, revealing a gelation temperature inversely related to the poloxamer concentration [31]. In the case of F127 20% *w*/*w*, the T_sol-gel_ was 20.6 °C, thus suitable for PG handling and administration on skin and mucosae. Approaching body temperature, the transition to gel should allow us to maintain the formulation in contact with HSV-1 sores.

PLO was obtained by mixing an organic phase, based on PC/IPP, with an aqueous phase, constituted of PG (F127 20, or 25%, *w*/*w*). Different approaches were considered, such as the alternative pouring of PC/IPP in PG or PG in PC/IPP using different volume ratios (Table 1). The addition of PC/IPP solution (PC 15.6%, *w*/*w*) to PG 25% (*w*/*w*) using volume ratios between the organic and water phases of 2:1, 1:1, or 1:2 (PLO-1, PLO-2, and PLO-6, respectively) resulted in unstable systems undergoing phase separation within a few minutes. A further increase in the aqueous phase, corresponding to a 17.8% *w*/*w* final concentration of F127, enabled us to obtain a stiffer homogeneous gel (PLO-7). As a second approach, PG (20%, *w*/*w*) was added under stirring to PC/IPP (200, 100, 50 mM) in a ratio by volume 70:30, resulting in homogeneous opaque thick gels (PLO-3, PLO-4, and PLO-5). PLO-3 and PLO-7 were selected on the basis of the highest PC content (being 4.68 and 4.4%, as reported in Table 2) and the possibly greater suitability in solubilizing MG while promoting its transdermal delivery. The compositions of the selected LO, PG, and PLO formulations are reported in Table 2.

### 3.2. Characterization of Semisolid Formulations

#### 3.2.1. Rheological Study

The consistency of formulations is one of the key features for their application on mucosae or skin. In this regard, a rheological study was conducted on LO, PG, PLO-3, and PLO-7. Indeed, the gel viscosity plays a pivotal role in drug permeation control [43]. Table 3 reports the consistency (K) and flow (n) indices determined at 20 °C from a fit of the gel flow curves (the shear stress versus shear rate). The flow index values of all gels were below unity, indicating their behavior as shear-thinning fluids since their viscosity decreases by increasing the shear rate (Figure 1). This behavior is typical of polymer solutions, particularly of Pluronic gels [23]. With regard to gel consistency, LO and PLO-3 displayed the lowest K indices, with similar values; the K index of PG was intermediate; and PLO-7 K index was the highest. The consistency of PG can be ascribed to its thermo-reversible gelling behavior. Indeed, at 20 °C, PG behaves as a viscous solution and starts its transition to a gel (20.6 °C) [31]. As previously found by other authors [25], in the case of PLO, the presence of PC increased the gel strength with respect to PG, resulting in the extensive formation of a network-like structure with higher viscosity. It can be hypothesized that POE groups of F-127 could interact with the phosphate groups of neighboring PCs as well as with water, finally leading to a complex thick three-dimensional structure [44]. Notably, PLO-7 behavior was surprising, being characterized by a consistency 7-fold higher with respect to PLO-3. This difference should be mainly related to the higher F127 concentration (17.8% vs. 14%) in PLO-7 with respect to PLO-3; indeed, an increase in the polymer concentration led to an increase in the gel viscosity (Table 3). In the case of PLO-7, a more compact supramolecular architecture is suggested, possibly due to hydrophobic interactions between the copolymer chains as well as to the structural network formed between F127 chains and the worm-like micelles formed by PC in the presence of water [44]. A peculiar character was also displayed by LO [19]. Indeed, LO behaved as a Newtonian fluid up to ~5 s^−1^ at 20 °C, exhibiting a constant viscosity, while under a further shear rate increment, LO became a shear-thinning fluid, displaying a viscosity decrease. This behavior could be attributed to the occurrence of a structural transition in the three-dimensional network of entangled polymer-like micelles under an applied flow field, as described by some authors [30,48]. Particularly, in LO, a shear rate increment over ~5 s^−1^ is supposed to induce a cylindrical aggregate alignment along the flow direction, leading to a disentanglement of the micellar aggregates [48].

To compare LO with the other gels, the consistency index K and flow index were computed at shear rate above 10 s^−1^, corresponding to the LO shear thinning part (Table 3).

The PLO-3 and PLO-7 T_sol-gel_ values were ≈ 18 and 13 °C, respectively. Since LO, PG, and PLO-3 were characterized by lower consistency with respect to PLO-7 and were thus more suitable for application on HSV-1 sores, they were further evaluated at 35 °C in view of their cutaneous administration (Figure 2). Notably, PLO-3 exhibited the same viscosity at 20 or 37 °C. The viscosity of PG was slightly higher due to its gel status being far above T_sol-gel_ while LO viscosity was significantly lower at 37 °C. Therefore, PLO-3 and PG are well-suited for mucosal or cutaneous application. Indeed, their shear thinning character enables us to easily handle and apply them on HSV-1 sores, forming a depot suitable for releasing MG at low shear rates while remaining rheologically stable at skin temperature.

#### 3.2.2. Spreadability Study

The spreadability of LO, PG, PLO-3, and PLO-7 was studied in order to obtain information on their application on mucosae and skin (Table 3). Indeed, spreadability can affect (i) gel extrudability from the package, (ii) the capability to cover mucosa or a skin area, (iii) patient compliance, and (iv) dosage transfer, thus influencing the gel therapeutic efficacy [34]. LO and PG were found to be more spreadable with respect to PLO formulations. Particularly, PLO-7 was not very spreadable, in agreement with rheological data, suggesting that PLO-7 was less suitable for mucosal application, being too stiff at ambient temperature (20 °C). For this reason, PLO-7 was not considered for further studies.

#### 3.2.3. Leakage and Adhesion Tests

The leakage potential of gels was compared to mimic in vitro their application on oral mucosa, with special regard to the lips or skin surface. The formulation running distance over the vertical plane reflects the leakage, which should be minimal to ensure prolonged action [34]. As shown in Figure 3 and reported in Table 4, LO ran the longest distance in 10 s, followed by PG and PLO-3. In this latter case, the gel maintained its position on the slide 1 h after its placement.

The capability of a material to adhere to a mucosal surface can be defined as adhesion. In order to achieve retention of a pharmaceutical dosage form on mucosae, a high adhesion is required [34,40]. The adhesions of LO, PG, and PLO-3 were evaluated in vitro by comparing their residence time on slides immersed in phosphate buffer, pH 7.4, at 32–35 °C in order to mimic the temperature of the lips and the skin (Figure 4). In the case of LO, the red area occupied on the slide underwent a 40% increase already after 10 min from application. After 1 h, the LO red area was undetectable, since it covered the whole slide and spilled out, floating in the phosphate buffer. On the other hand, the areas of PG and PLO-3 underwent slight increases at 10 min, hardly reaching increases of 20% or 5%, respectively, after 1 h. Therefore, PLO-3 displayed a more lasting adhesion with respect to PG, whilst LO adhesion was negligible, in agreement with the spreadability and leakage findings.

The skin adhesions of LO, PLO-3, and PG were further evaluated by applying the formulations onto the forearm of a volunteer rotated outward and by observing their shift after rotating the forearm inward. As depicted in Appendix A, PG and PLO-3 maintained their position even 10 min after vertical shift of the forearm, whilst LO descended immediately after the inward rotation. These observations confirmed the in vitro adhesion test findings and suggested the lack of suitability of LO for cutaneous application and treatment of HSV-1 sores.

### 3.3. In Vivo Comparative Irritation Test

In order to in vivo evaluate if LO, PG, and PLO-3 could induce irritative reaction when applied onto the skin, a patch test was performed on 20 health volunteers. The number of irritative reactions turned up at 15 min and 24 h after removal of the Finn Chamber was recorded and expressed as irritation indexes (Table 4). Noteworthily, LO induced a negligible reaction, while PG and PLO did not induce any kind of reaction. All gels can be classified as not irritating if applied onto the human skin. 

### 3.4. Preparation of MG-Containing Semisolid Forms

MG-containing gels were produced by alternatively solubilizing the drug in PC/IPP solution in the case of LO or in F127 solution in the case of PG and PLO-3 (Table 1). The MG-containing gels appeared yellowish and homogeneous. The amount of MG loaded in the gels was quantified by dilution, disaggregation and HPLC. In all cases, MG recovery corresponded to the total amount weighed for the gel production, suggesting that the preparation procedure performed by cold methods avoided drug degradation due to high temperatures. The MG residual content in the different gels stored in the light for 3 months was evaluated as a percentage of the total amount of drug used for the preparation and compared with the MG content in SOL-MG. As shown in Figure 5, all gels demonstrated a remarkable efficacy in controlling MG stability. Indeed after 3 months, the MG amount was above 97% for PG-MG and PLO-3-MG, or 93% in the case of LO-MG, while MG content underwent a 15% decrease in the case of SOL-MG, suggesting that semisolid forms were able to control the chemical stability of MG. The macroscopic aspect of all of the gels did not change with time, appearing devoid of phase separation at least 6 months after production. 

### 3.5. In Vitro Diffusion Kinetics of MG

The influence of the vehicles on MG diffusion was evaluated by comparing LO-MG, PG-MG, PLO-3-MG, and SOL-MG using Franz cell experiments and was associated with a synthetic polymeric membrane constituted of polytetrafluoroethylene. Even though animal or human skin provide more predictable permeation results with respect to synthetic membranes, their use implies many drawbacks related to their scarce availability, ethical restrictions, variable results, and risks of contamination. On the other hand, synthetic membranes are recommended by the FDA in Franz cell tests to compare the performances of topical forms or for quality controls, being chemically inert and commercially available, and providing highly reproducible results [31,35].

As reported in Figure 6a,b, as expected, all semisolid forms enabled us to control MG diffusion with respect to SOL-MG. The MG diffusion parameters are reported in Table 5.

The D values trend, obtained by considering MG concentration in the different formulations, was PLO-3-MG < PG-MG < LO-MG < SOL-MG. Particularly, in the case of SOL-MG, the drug diffusion was the fastest within 3 h; afterwards, it reached a plateau, probably due to the achievement of a MG concentration balance between the donor and the receiving phases of the Franz cell.

Notably, in the case of PLO-3-MG, considered a hybrid form between LO-MG and PG-MG, the amount of MG diffused after 7 h was 12-fold lower compared with that diffused from SOL-MG, while the diffusion was 8-fold slower, suggesting that both PC/IPP and F-127 networks contribute to considerably restrained diffusion of the drug with respect to the other forms. On the other hand, the F-127 micellar network in the hydrogel PG-MG was able to reduce MG diffusion by almost 5-fold with respect to the ethanolic solution of the drug. In the case of LO-MG, the supramolecular organization of PC as well as the presence of IPP controlled MG diffusion with respect to SOL-MG; nonetheless, its lower viscosity resulted in a faster diffusion of the drug with respect to PG-MG and PLO-3-MG.

### 3.6. In Vitro Antiviral Activity

A plaque reduction assay was used to evaluate the in vitro anti-HSV-1 activities of MG-containing formulations. Particularly, different protocols were employed based on formulation–cell contact followed by virus inoculation (pre-treatment), the addition of formulations to infected cells after virus removal (post-infection), and the formulation-on-virus direct action (virucidal assay). The formulations were added during different phases of infection to determine the time and mode of action. A preliminary study allowed us to select 50 ug/mL MG as the concentration, being efficacious and not toxic to Vero cells up to 500 ug/mL, as described by Rechenchoski et al. [12]. The percent reduction was calculated relative to the amount of viral control in the absence of formulations. The cell pre-treatment with PG-MG, PLO-3-MG, LO-MG, and SOL-MG before viral infection or the addition of formulations during the adsorption phase for 1 h did not reduce the plaque formation. On the other hand, the formulations exhibited virucidal activity when incubated with the virus before cell infection. Indeed after 1 h of incubation at 35 °C and subsequent titration on the cell monolayer, a 40% reduction in viral growth was observed for all formulations, except for LO-MG, which did not show any virucidal activity (Table 6). It can be hypothesized that the oleaginous nature of LO resulted in phase separation in contact with the aqueous medium, hampering the contact between the loaded MG and the HSV-1 KOS strain. Conversely, the virucidal activity of PG-MG and PLO-3-MG was further evaluated, prolonging direct contact of the formulations with the viral HSV-1 KOS strain at 35 °C to 6 h before infection of the cell monolayer.

Noteworthily, after 6 h, the plaque reduction was always above 93%, as reported in Table 6 and depicted in Figure 7. This result suggests that MG has the ability to inhibit viral growth, possibly acting on the viral envelope and thus reducing the adsorption and penetration capacity of the virus. Particularly, in the case of PLO-3-MG, the virucidal effect was the highest (viral growth 2%), followed by PG-MG and SOL-MG.

## 4. Conclusions

This investigation enabled us to select smart gels for MG delivery on skin and mucosa. Particularly, the formulative study evidenced the suitability of PG and PLO-3 over LO due to their rheological behavior, spreadability, and adhesion. All gels were able to solubilize MG and to maintain drug stability over 3 months. The in vitro Franz cell diffusion test confirmed the capability of PG-MG and PLO-3-MG to control MG diffusion with respect to the low viscous LO-MG. Notably, both PG-MG and PLO-3-MG evidenced a virucidal activity against HSV-1, particularly evident when incubating the formulations with the virus for 6 h before cell monolayer infection. The absence of antiviral activity observed 1 h before infection concurrently with the virus inoculum or 1 h post-infection could be attributed to the short contact time of formulations with the cells before or after infection. These encouraging results suggest the possibility to employ MG-containing gels as adjuvants to slow down HSV-1 infection. However, further studies will be required to better understand the kinetics as well as the antiviral activity mechanism of MG-loading gels. Particularly, to investigate the transdermal effect of the gels, epidermal sheets of murine skin [49] will be possibly employed to study MG permeation and as ex vivo models of HSV-1 infection.

## Figures and Tables

**Figure 1 pharmaceutics-13-01323-f001:**
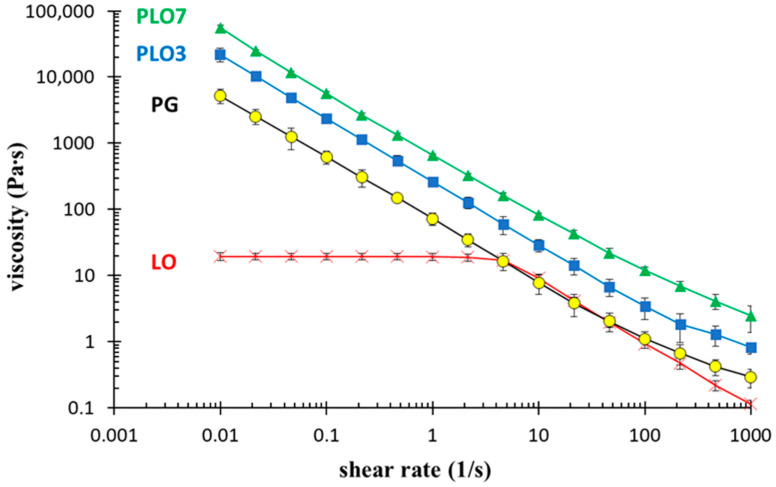
Rheological behavior of LO, PG, PLO-3, and PLO-7 as a function of shear rate measured at 20 °C.

**Figure 2 pharmaceutics-13-01323-f002:**
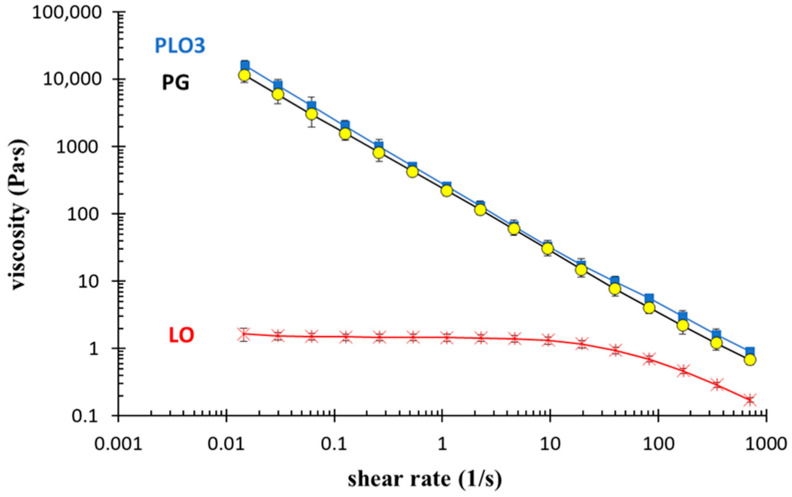
Rheological behavior of LO, PG, and PLO-3 as a function of shear rate measured at 35 °C.

**Figure 3 pharmaceutics-13-01323-f003:**
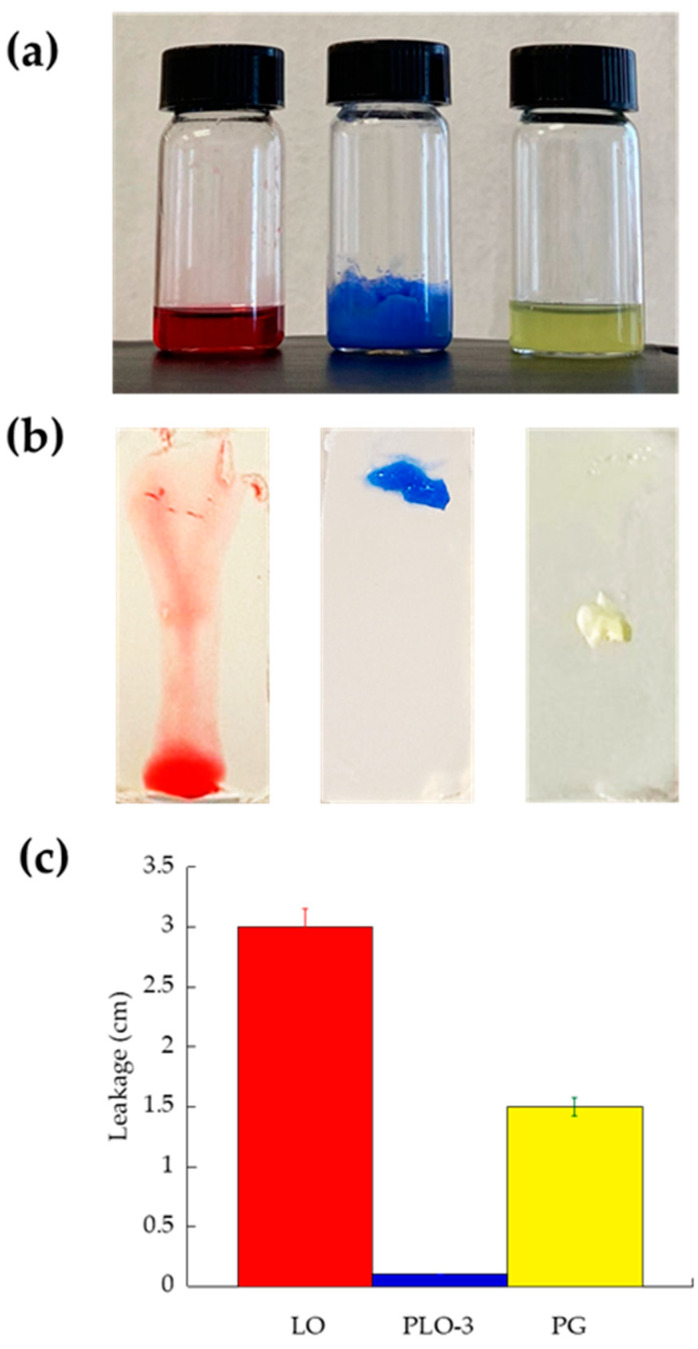
Comparative leakage test performed on LO (red), PLO-3 (blue), and PG (yellow) gels colored for imaging (**a**) and placed on pH 7.4 agar slides (**b**) at 32–35 °C. The leakage distance was measured 10 s after gel application (**c**).

**Figure 4 pharmaceutics-13-01323-f004:**
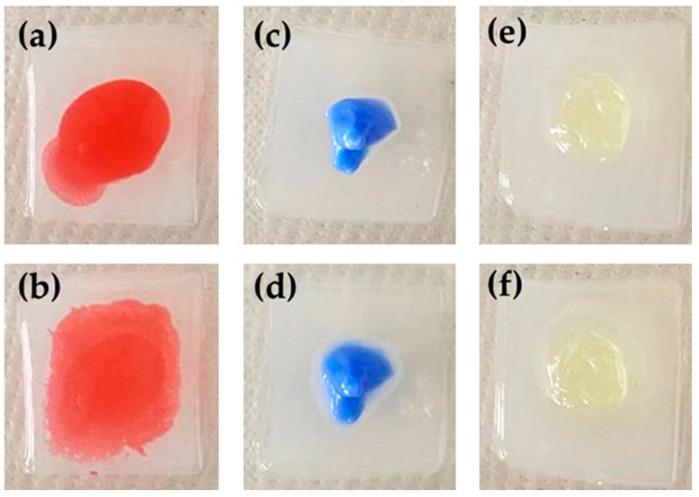
Comparative adhesion test performed on pH 7.4 agar plates immersed in 10 mL of phosphate buffer, pH 7.4, at 32–35 °C. The images were taken 1 (**a**,**c**,**e**) or 10 (**b**,**d**,**f**) min after applications of LO (**a**,**b**) PLO-3 (**c**,**d**), or PG (**e**,**f**).

**Figure 5 pharmaceutics-13-01323-f005:**
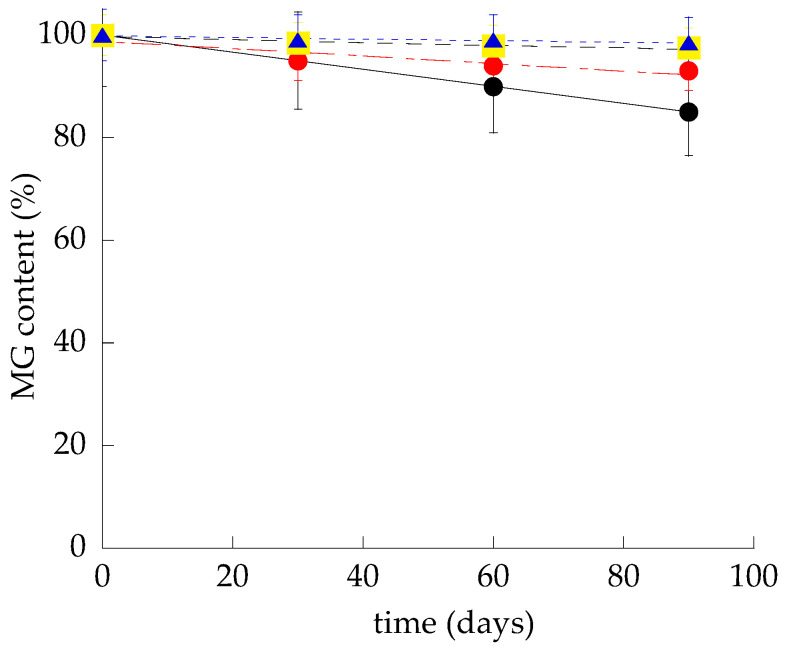
MG stability in SOL-MG (•), LO-MG (•), PG-MG (■), and PLO-3-MG (▲) samples under 3 months of storage. The data are the mean of three independent experiments.

**Figure 6 pharmaceutics-13-01323-f006:**
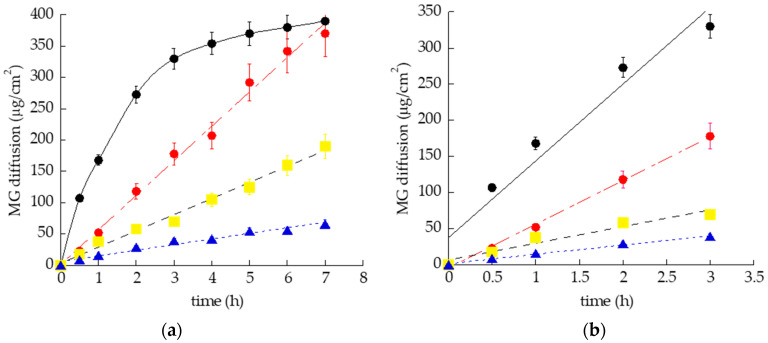
MG diffusion kinetics from SOL-MG (•), LO-MG (•), PG-MG (■), and PLO-3-MG (▲), as determined by the Franz cell experiment. The experiments were conducted for 8 h (**a**). Panel (**b**) refers to the linear part of the kinetics (0–3h). The data are the mean of six independent experiments.

**Figure 7 pharmaceutics-13-01323-f007:**
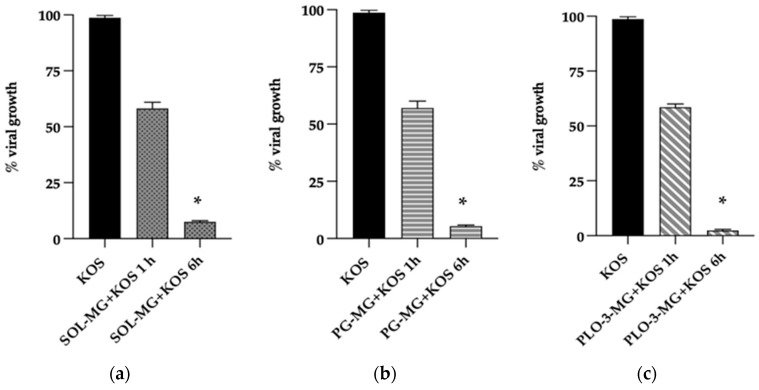
Virucidal effect of SOL-MG (**a**), PG-MG (**b**), and PLO-3-MG (**c**) against the HSV-1 KOS strain (1 × 10^5^ pfu/mL). The virus was incubated for 1 and 6 h at 35 °C with 50 mg/mL of MG loaded in the different gels (**b**,**c**) or in SOL-MG (**a**). The data represent the percentage of viral growth after 1 or 6 h of incubation with the formulations. The values were obtained by comparing the results with those of the control. The data represent the mean of three independent experiments; * *p*-values < 0.05.

**Table 1 pharmaceutics-13-01323-t001:** Composition of PLO formulations.

PLO Code	O ^1^ Phase (% *w*/*w*)		W ^2^ Phase (% *w*/*w*)		O/W Phase Ratio (*v*/*v*)	Notes
	PC ^3^	IPP ^4^	F127 ^5^	Water		
PLO-1	15.6	84.4	25	75	2:1	P.S.^6^
PLO-2	15.6	84.4	25	75	1:1	P.S. ^6^
PLO-3	15.6	84.4	20	80	1:2.3	H.F.^7^
PLO-4	7.8	92.2	20	80	1:2.3	H.F.^7^
PLO-5	3.9	96.1	20	80	1:2.3	H.F.^7^
PLO-6	15.6	84.4	25	75	1:2	P.S.^6^
PLO-7	15.6	84.4	25	75	1:2.6	H.F.

^1^ organic; ^2^ aqueous; ^3^ soy phosphatidylcholine; ^4^ Pluronic F-127; ^5^ isopropyl palmitate; ^6^ phase separation; and ^7^ homogeneous form.

**Table 2 pharmaceutics-13-01323-t002:** Composition of LO, PG, and PLO formulations selected for this study.

Formulation Code	PC ^1^ % *w*/*w*	F-127 ^2^ % *w*/*w*	IPP ^3^ % *w*/*w*	Ethanol % *w*/*w*	Water % *w*/*w*	MG ^4^ % *w*/*w*
LO	15.6	-	82.96	-	1.44	-
PG	-	20.0	-	-	80.0	-
PLO-3	4.68	14.0	25.32	-	56.0	-
PLO-7	4.40	17.8	28.13	-	49.67	-
LO-MG	15.60	-	82.89	-	1.44	0.07
PG-MG	-	20.0	-	-	79.93	0.07
PLO-3-MG	4.68	14.0	25.32	-	55.93	0.07
SOL-MG	-	-	-	69.93	30.0	0.07

^1^ soy phosphatidylcholine; ^2^ Pluronic F-127; ^3^ isopropyl palmitate; and ^4^ mangiferin.

**Table 3 pharmaceutics-13-01323-t003:** Rheological parameters and spreadability of the indicated gels measured at 20 °C. Consistency index K and flow index were computed at shear rate above 10 s^−1^ for comparison.

Formulation Code	K Index ^1^ (Pa s^n^) *	Flow Index	Viscosity (Pa s at 1000 s^−1^)	Spreadability (g cm/s)
LO	80.0 ± 3.0	0.05 ± 0.01	0.11 ± 0.02	15.0 ± 0.7
PG	177.0 ± 48.0	0.10 ± 0.01	0.29 ± 0.10	15.2 ± 0.2
PLO-3	91.0 ± 11.0	0.31 ± 0.01	0.81 ± 0.17	9.2 ± 0.4
PLO-7	657.0 ± 79.0	0.10 ± 0.03	2.45 ± 1.07	5.3 ± 0.3

^1^ consistency index; * n: flow index; data are the mean of three independent experiments.

**Table 4 pharmaceutics-13-01323-t004:** Leakage and irritation index of the indicated formulations.

Formulation	Leakage ^1^ (cm)	Irritation Index (Mean) ^2^	
		0.25 h	24 h
LO	3.0 ± 0.3	0.1	0.1
PG	1.5 ± 0.1	0.0	0.0
PLO-3	0.0 ± 0.0	0.0	0.0

^1^ running distance along the slide, ^2^ n = 20; mean data of three independent experiments.

**Table 5 pharmaceutics-13-01323-t005:** Diffusion parameters of the indicated formulations.

Parameter	LO-MG	PG-MG	PLO 3-MG	SOL-MG
F ^1^ ± s.d. (μg/cm^2^/h)	60.5 ± 6.8	23.0 ± 2.4	12.9 ± 1.1	106.2 ± 12
MG (mg/mL)	0.7	0.7	0.7	0.7
D ^2^ ± s.d. (cm/h) × 10^−3^	86.5 ± 9.7	32.9 ± 3.4	18.42 ± 1.6	151.7 ± 17
Q_6_ ^3^ ± s.d. (cm/h) × 10^−3^	370.0 ± 38	66.0 ± 5.8	32.0 ± 1.8	390.0 ± 4.1
Reduction ratio ^4^	1.75	4.6	8.23	-

^1^ flux; ^2^ diffusion coefficient; ^3^ Q_6_ diffused after 7 h; ^4^ reduction in diffusion with respect to SOL-MG; data are the mean of six independent Franz cell experiments.

**Table 6 pharmaceutics-13-01323-t006:** Percentage values of plaque reduction after direct contact between the indicated formulations and HSV-1 KOS strain 1 × 10^5^ PFU/mL at 35 °C.

Formulation	Plaque Reduction (%) 1 h	Plaque Reduction (%) 6 h
SOL-MG	40.0 ± 1.70	93.0 ± 0.74
PG-MG	40.0 ± 1.78	94.0 ± 0.74
PLO-3-MG	40.0 ± 1.00	98.0 ± 0.62
LO-MG	-	-

## Data Availability

The data presented in this study are available from the corresponding author upon request. The data are not publicly available due to privacy restrictions.

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
