# Peer review of "Mangiferin-Loaded Smart Gels for HSV-1 Treatment"

_pharmaceutics, 2021, doi:10.3390/pharmaceutics13091323_

Round 1

Reviewer 1 Report

The in vitro and in vivo antiviral activity of mangiferin has already been studied, however, the present study investigated the application of the compound in different formulations, including initial studies in humans. The results are unprecedented and relevant to the scientific community. The methodology and results were well described. Therefore, I consider the article accepted for publication.

Author Response

We thank the reviewer for his kind comment.

Reviewer 2 Report

In this paper, drug delivery systems were developed to carry and deliver the low-soluble drug, mangiferin (MG), for herpes infection treatment. Three different systems were tested:  Lecithin organogel (LO), pluronic hydrogel (PG), and pluronic lecithin organogel (PLO-3 and PLO-7). All the formulations were able to solubilize the drug MG.  PLO-7 was discarded, due to the highest consistency and low spreadability. PLO-3 presented the lowest leakage and the highest adhesion, either on agar slides or human skin. PG and PLO-3 were able to control the MG release, and both showed a virucidal activity against HSV-1. The study presents encouraging results to use MG-loaded gels to treat resistant herpes infection. The manuscript could be of potential interest to the readers and could be suggested for publication in Pharmaceuticals after revision.  

In the title, the authors claim that the gels are smart, but the explanation of this feature is unclear in the Introduction. 

In the Materials and Methods, in the item 2.4, the authors should define the samples PLO-3 and PLO-7, considering they are cited in the next item, 2.5. 

In the line 395, how was the Tsol-gel determined for PLO-3 and PLO-7? 

The authors state that delivery systems were developed for topical application, including skin and mucosa. In the adhesion tests, an environment pH of 7.4 was used. The authors should mention which biofluid is being simulated. For what type of mucosa (e.g., vaginal, oral) would the formulations be?  

In the Tables 3, 5 and 6 the significant figures of the numbers should be reviewed.  

In the Table 5, the index 4 is missing. 

In the line 503, the authors described a diffusion 18-fold lower for PLO-3 instead of 8-fold, as indicated in the Table 5. 

During the evaluation of the MG content in the formulations after 30 days, is it possible to identify the possible degradation products? What is the specificity of the employed analytical method? 

Author Response

Point 1. In the title, the authors claim that the gels are smart, but the explanation of this feature is unclear in the Introduction. 

Response 1. We thank the reviewer for this suggestion, “smart” was yet specified at line 76, furthermore the use of smart in the revised version has been claimed at line 73 and added at line 90.

Point 2. In the Materials and Methods, in the item 2.4, the authors should define the samples PLO-3 and PLO-7, considering they are cited in the next item, 2.5. 

Response 2. We thank the reviewer for this suggestion, we changed line 129-130 accordingly.

Point 3. In the line 395, how was the Tsol-gel determined for PLO-3 and PLO-7? 

Response 3. Tsol-gel was determined at the G'-G'' crossover (G': elastic modulus, G'': viscous modulus) from temperature ramp experiments in oscillation mode and in the linear regime. Temperature ramps were taken from 5 °C to 50 °C with a temperature rate of 1 °C/min, controlled by a Peltier plate. A 2-min conditioning time at 5 °C was applied before starting the experiments. Measurements were conducted at least thrice for each sample. The information has been inserted at lines 139-143.

Point 4. The authors state that delivery systems were developed for topical application, including skin and mucosa. In the adhesion tests, an environment pH of 7.4 was used. The authors should mention which biofluid is being simulated. For what type of mucosa (e.g., vaginal, oral) would the formulations be?  

Response 4. The formulations were designed for oral mucosa and particularly for the lips. The pH 7.4 should mimic saliva. This is specified in the revised manuscript at lines 177-178,418,419,433,444.

Point 5. In the Tables 3, 5 and 6 the significant figures of the numbers should be reviewed.  

Response 5. In the Tables 3, 5 and 6 the significant figures of the numbers have been revised accordingly.  

Point 6. In the Table 5, the index 4 is missing. 

Response 6. The index has been inserted accordingly in Table 5.

Point 7. In the line 503, the authors described a diffusion 18-fold lower for PLO-3 instead of 8-fold, as indicated in the Table 5. 

Response 7. We thank the reviewer for this suggestion, the mistake has been corrected at line 499.

Point 8. During the evaluation of the MG content in the formulations after 30 days, is it possible to identify the possible degradation products? What is the specificity of the employed analytical method? 

Response 8. Degradation products were not identified since the HPLC peaks were not different after 30 days with respect to time 0. The method was validated for linearity (R2 = 0.994), repeatability (relative standard deviation <0.05 %, n = 6 injections) and limit of quantification (0.05 μg/ml). This is specified at lines 214-216.

Reviewer 3 Report

The authors investigated smart gels that include mangiferin, a natural flavonoid glycoside, for HSV-1 treatment. Smart gels receive increasing attention as convenient application form of many natural and synthetic drugs to treat various ailments. The authors prepared and tested several types of gels or sols, evaluated their effects, and summarized the results.

The experiments were relatively well planned, and the investigation was logically performed. However, this investigation should be extended at least for the transdermal effect, if any of those formulations should have a practical impact. I strongly recommend to include such a study into this manuscript now, instead of promissing to perform additional experiment that might be never done in the future.

From the formal point of view, I found several errors that should be corrected before the manuscript can be accepted:

Table 2: The data in the third column from left seem to be incorrectly written, and they should be edited and corrected.

Figure 4: The text in the note under the figure is not correct. It seems that the images taken in the minute 1, are the images (a), (c) and (e), while the authors state images (a-e). Similarly, images taken in the minute 10 cannot be (b-f), because this gives no sense. They are very probably images (b), (d) and (f). The corresponding text in the whole paragraph should be checked and revised.

Line 469: Possibly missing word(s).

Table 6: The data in the second column should be adjusted.

Figure 7: The notes under the figure 7 refer to the figures (a), (b) and (c). However, those marks are not shown at any of the pictures of the Figure 7. Please, revise and correct the data.

Summary: I recommend to include the missing data to support practical applicability of the investigated gels / sols. Therefore, I recommend to reconsider this manuscript after completing the missing data.

Author Response

Point 1: However, this investigation should be extended at least for the transdermal effect, if any of those formulations should have a practical impact. I strongly recommend to include such a study into this manuscript now, instead of promissing to perform additional experiment that might be never done in the future.

Response 1: We thank the reviewer for his comment and suggestion. The transdermal effect will be evaluated in the next manuscript we are going to prepare, thus additional experiments will be surely performed in future using epidermal sheets of murine skin, as well as human skin explants.

Unfortunately we cannot include them in the present manuscript since we cannot perform the experiments within 10 days. Particularly we should wait for murine skin because of ethical reasons and obviously samples of human skin are very difficult to obtain. Anyway, we planned to perform those experiments in autumn 2021 thanks to the collaboration with our colleague who promised to get skin samples from September.

However we would like to underline that a technological study is always stronlgy required to get the composition of a new pharmaceutical form, in this respect the aim of this formulative study was the selection of semisolid formulation for MG delivery to treat HIV-1. Thanks to the encouraging results we obtained, we are strongly interested to proceed with further studies.

From the formal point of view, I found several errors that should be corrected before the manuscript can be accepted:

Point 2: Table 2: The data in the third column from left seem to be incorrectly written, and they should be edited and corrected.

Response 2: We thank the reviewer but we controlled the data calculating again the percentage composition and we confirm that they are correct. Please consider that Table 1 reports the relative F127 concentration in the water phase, specifying the O/W phase volume ratio in PLO gels. Instead in Table 2 for all gels the absolute F127 concentration is reported. Anyway we emended some errors we found in the water column of Table 2 and in Table 1.

Point 3: Figure 4: The text in the note under the figure is not correct. It seems that the images taken in the minute 1, are the images (a), (c) and (e), while the authors state images (a-e). Similarly, images taken in the minute 10 cannot be (b-f), because this gives no sense. They are very probably images (b), (d) and (f). The corresponding text in the whole paragraph should be checked and revised.

Response 3: We thank the reviewer, accordingly we emended Figure 4 caption. The whole paragraph was checked but it did not need to be changed.

Point 4: Line 469: Possibly missing word(s).

Response 4: The missing words have been inserted accordingly at lines 466-469.

Point 5: Table 6: The data in the second column should be adjusted.

Response 5: The data have been corrected.

Point 6: Figure 7: The notes under the figure 7 refer to the figures (a), (b) and (c). However, those marks are not shown at any of the pictures of the Figure 7. Please, revise and correct the data.

Response 6: Sorry for the inconvenience, there has been a problem in the formatting of the manuscript and Figure 7 in the template. We emended it in the revised version indicating a, b and c panels.

Round 2

Reviewer 3 Report

The authors answered all my questions and made adequate modifications in the text to achieve improvement of the manuscript.

I understand that to perform additional experiments within several days given for revision, is not possible.

Therefore, I recommend to accept this manuscript for publication in its current revised form.